# GAME FORMS OF VAN DER WAERDEN GAMES

**Dobritsyn Michail Dmitrievich**
HSE University
Moscow, Russia

## ABSTRACT

Finding winning strategies in different games is important area of research. In this paper we focus on positional games only, i.e. games where players alternately color previously uncolored verticies and players goal is to color one of the winning sets. Despite simple definition [Slany (1999)] showed that even this problem in PSPACE complete. It means that practical implementations of strategies are limited to heuristics and AI. However, purely mathematical strategies are still important, as they can be used to compare advanced methods to them (as a baseline) and because they require lesser amount computation. In this paper we discuss strategies for both Maker and Breaker which guarantee win under certain condition, meaning they can be used as starting points for programming strategies for concrete positional games that might include heuristics and AI.

From a mathematical point of view we study natural game version (maker-breaker game) of popular branch of combinatorics Ramsey type — Ramsey theory. Those games can be used to study Ramsey type problems, which is done in [Slany (1999)] [Ai et al. (2025)] [Brown et al. (2020)]. Here we talk about game forms of two families of problems (including open ones) and discuss how the strategies work in this case.

## 1 INTRODUCION

Informally, Ramsey theory studies the problems of type "Does the large size of arbitrary structure guaranties existence of specific regular substructure". In this paper we focus only on one popular subclass of Ramsey type problems which state "Given predicate $P$ and sequence of sets $S_1 \subset S_2 \subset \ldots$ is it true that for any number of colors $c \in \mathbb{N}$, exists such $N \in \mathbb{N}$ so that for any coloring of $S_N$ in $c$ colors, exists monochromatic subset of $S_N$ satisfying a property $P$". Monochromatic subset is a subset which elements are colored in the same color.

This is famous branch of combinatorics with famous results such as Van der Waerden theorem [van der Waerden (1927)], Szemeredi theorem [Szemerédi (1974)], and theorem proven by Terence Tao on arithmetic progression consisting of prime numbers [Tao (2008)].

One of the new approaches to this type of problems is to define games associated with those problems and study them instead. Given a problem of a Ramsey type, one can construct a positional game defined as follows "Two players alternately pick previously unpicked elements of $S_N$. First player picks one element per turn, other picks $c - 1$ elements per move. The game ends when all vertices are picked. First player (maker) wins if he occupies a subset of $S$ that satisfies property $P$, second player (breaker) wins otherwise.". Our goal is to understand if we can determine which player wins for a large enough value of $N$.

This approach was introduced and studied by Joseph Beck in [Beck (1981)] for Van der Waerden game for 2 colors and was done for $c$ colors, but the winning condition were asymptotic. In this work we use method developed by Joseph Beck [Beck (1981)] [Beck (1982)] to give explicit formulas for the Van der Waerden Game, its $n$-dimensional analogy and Hales-Jewwet game. We then take two families of Ramsey type problems, construct associated games and solve them (we find which player has a winning strategy).

## 2 TERMINOLOGY

Our positional game can be defined by set of all vertices $S_n$ and set $\mathcal{F}$ of all subsets of $S_n$ satisfying property $P$. Such structure of set $\mathcal{F}$ of subsets of $S_n$ is called a hypergraph, elements of $\mathcal{F}$ are called edges and elements of $S_n$ are called vertices. Let's denote $\mathcal{V}(\mathcal{F})$ the set of all vertices in hypergraph $\mathcal{F}$.

In this and next section we deal with finite hypergraphs ($|\mathcal{F}| < \infty \wedge \forall S \in \mathcal{F} \wedge |S| < \infty$).

The hypergraph is called $r$-uniform if every its edge has exactly $r$ vertices. For vertices $v_1, v_2$ let's denote $d_{v_1 v_2}(\mathcal{F}) := |\{S \in \mathcal{F} \mid v_1 \in S \wedge v_2 \in S\}|$ — number of edges that contain both $v_1$ and $v_2$. Let's also denote $d_2(\mathcal{F}) = \max\limits_{v1, v2 \in \mathcal{V}(\mathcal{F}))} d_{v_1 v_2}(\mathcal{F})$ — maximum number of edges intersecting by at least two vertices.

The game corresponding to the hypergraph $\mathcal{F}$ and number of colors $c \in \mathbb{N}$ is denoted as $G(\mathcal{F}, c)$.

## 3 GENERAL THEOREMS [BECK (1981)] [BECK (1982)]

We shall prove two theorems which provide sufficient conditions for Maker to have a winning strategy and Breaker to have a winning strategy based only on relations between $|\mathcal{F}|$, $|\mathcal{V}(\mathcal{F})|$, $r$ and $c$.

**Theorem 1.** *If $\mathcal{F}$ is a $r$-uniform hypergraph and $c \in \mathbb{N}$ satisfy $|\mathcal{F}| < c^{r-1}$, then Breaker has a winning strategy in the $G(\mathcal{F}, c)$.*

**Theorem 2.** *If $\mathcal{F}$ is a $r$-uniform hypergraph and $c \in \mathbb{N}$ satisfy $\frac{|\mathcal{F}|}{|\mathcal{V}(\mathcal{F}))|} > (c-1)^2 c^{r-3} d_2(\mathcal{F})$ than the Maker has a winning strategy in $G(\mathcal{F}, c)$.*

### 3.1 DEFINITION OF A POTENTIAL AND IT'S PROPERTIES

Suppose on $m$-th move of the game first player have picked vertices $\{x_1 \ldots x_m\} = X$ and second player have picked $\{y_1 \ldots y_{(c-1)m}\} = Y$. Let's denote potential of this position. First step is to define potential of one edge $S \in \mathcal{F}$ as

$$P_{X,Y}(S) = \begin{cases} 0 & \text{if } S \cap Y \neq \emptyset \\ c^{-|\{S \backslash X\}|} & \text{otherwise} \end{cases}$$

Intuitively, $P_{X,Y}(S)$ is a probability that all vertexes of edge $S$ will be picked by first player in the end of the game if all unpicked vertices would be randomly added to $X$ or $Y$ with probabilities $\frac{1}{c}$ and $\frac{c-1}{c}$ respectfully.

Two notable properties of this functions are: $P_{X,Y \cup Y'}(S) \leqslant P_{X,Y}(S)$ for all $X, Y, Y', S \in \mathcal{F}$ and if $S \cap (X' \cup Y') = \emptyset$ then $P_{X,Y}(S) = P_{X \cup X', Y}(S) = P_{X, Y \cup Y'}(S) = P_{X \cup X', Y \cup Y'}(S)$.

The potential of the state of the game $(X, Y)$ is the sum of potentials of all edges of the hypergraph $\mathcal{F}$:

$$P(X, Y) = \sum_{S \in \mathcal{F}} P_{X,Y}(S)$$

Intuitively, $P(X, Y)$ is a mathematical expectation of number of edges that would be completely picked by Maker in the end of the game if all unpicked vertices would be randomly added to $X$ or $Y$ with probabilities $\frac{1}{c}$ and $\frac{c-1}{c}$ respectfully.

#### 3.1.1 POTENTIAL AT THE BEGINNING OF THE GAME

In the begging of the game $X = \emptyset \wedge Y = \emptyset$, potential of each edge is $c^{-|S|} = c^{-r}$. Thus potential of the position $P(\emptyset, \emptyset) = \sum\limits_{S \in \mathcal{F}} c^{-r} = c^{-r} |\mathcal{F}|$

### 3.1.2 POTENTIAL IN THE END OF THE GAME

Useful non-trivial property of the potential is that by calculating potential of the position in the end of the game we can determine which player won.

Indeed, in the end of the game potential $P_{X,Y}(S)$ is 0, if at least one vertex of $S$ is picked by Breaker and $P_{X,Y}(S) = c^{S \setminus X} = c^0 = 1$ if all vertices are occupied by Maker. This means that in the end of the game potential is

$$
\begin{aligned}
P(X, Y) = \sum_{S \in \mathcal{F}} P_{X,Y}(S) &= \\
&= \sum_{S \in \mathcal{F} \wedge S \not\subset X} P_{X,Y}(S) + \sum_{S \in \mathcal{F} \wedge S \subset X} P_{X,Y}(S) = \\
&= \sum_{S \in \mathcal{F} \wedge S \not\subset X} 0 + \sum_{S \in \mathcal{F} \wedge S \subset X} 1 = \\
&= |\{S \in \mathcal{F} \mid S \subset X\}|
\end{aligned}
\tag{1}
$$

If Maker won (exist edge occupied by Maker), than the potential is at least 1, if Breaker won (none of the edges are completely occupied by Maker) potential is 0.

### 3.2 IDEA OF A PROOF

The idea of proof of sufficient conditions is to calculate the potential at the begging of the game, maximum speed of its change of the potential per one turn under certain strategies of the players and using that estimate the potential in the end of the game and thus a winner. Thus, selecting right strategy for a player can guaranty his win.

### 3.3 CHANGE OF POTENTIAL

Let's we define how much potential of the position would increase if Maker picks vertex $v$ in position $(X, Y)$ as $\Delta_M P_{X,Y}(v) = P(X \cup \{v\}, Y) - P(X, Y)$.

$$
\begin{aligned}
&\Delta_M P_{X,Y}(v) = \\
&= P(X \cup \{v\}, Y) - P(X, Y) = \\
&= \sum_{s \in \mathcal{F}} \left( P_{X \cup \{v\}, Y}(S) - P_{X,Y}(S) \right) = \\
&= \sum_{\substack{s \in \mathcal{F} \\ v \notin S}} \Big( \underbrace{P_{X \cup \{v\}, Y}(S)}_{\substack{= P_{X,Y}(S) \\ \text{because v not in S}}} - P_{X,Y}(S) \Big) + \sum_{\substack{s \in \mathcal{F} \\ v \in S}} \Big( \underbrace{P_{X \cup \{v\}, Y}(S)}_{\substack{= c P_{X,Y}(S) \\ \text{because v in S}}} - P_{X,Y}(S) \Big) = \\
&= 0 + \sum_{\substack{s \in \mathcal{F} \\ v \in S}} (c - 1) P_{X \cup \{v\}, Y}(S) = \\
&= (c - 1) \sum_{\substack{s \in \mathcal{F} \\ v \in S}} P_{X \cup \{v\}, Y}(S)
\end{aligned}
$$

Similary we define how much potential of the position would increase if Breaker picks vertex $v$ in position $(X, Y)$ as $\Delta_B P_{X,Y}(v) = P(X, Y \cup \{v\}) - P(X, Y)$.

$$\Delta_B P_{X,Y}(v) = \tag{2}$$

$$= P(X, Y \cup \{v\}) - P(X, Y) = \tag{3}$$

$$= \sum_{s \in \mathcal{F}} \left( P_{X, Y \cup \{v\}}(S) - P_{X,Y}(S) \right) = \tag{4}$$

$$= \sum_{\substack{s \in \mathcal{F} \\ v \notin S}} \left( \underbrace{P_{X, Y \cup \{v\}}(S)}_{\substack{= P_{X,Y}(S) \\ \text{because v not in S}}} - P_{X,Y}(S) \right) + \sum_{\substack{s \in \mathcal{F} \\ v \in S}} \left( \underbrace{P_{X, Y \cup \{v\}}(S)}_{\substack{= 0 \\ \text{because v in S}}} - P_{X,Y}(S) \right) = \tag{5}$$

$$= 0 + \sum_{\substack{s \in \mathcal{F} \\ v \in S}} -P_{X,Y}(S) = \tag{6}$$

$$= -\frac{1}{c-1} \Delta_M P_{X,Y}(v) \tag{7}$$

One notable property of $\Delta_B$ is

$$\Delta_B P_{X, Y \cup Y'}(v) \geqslant \Delta_B P_{X,Y}(v) \text{ for all } X, Y, Y' \in \mathcal{F} \text{ and } v \in \mathcal{V}(\mathcal{F}) \tag{8}$$

$$\Delta_M P_{X, Y \cup Y'}(v) \leqslant \Delta_M P_{X,Y}(v) \text{ for all } X, Y, Y' \in \mathcal{F} \text{ and } v \in \mathcal{V}(\mathcal{F}) \tag{9}$$

It is true because $\Delta_B P_{X, Y \cup Y'}(v) = -\sum_{\substack{s \in \mathcal{F} \\ v \in S}} P_{X, Y \cup Y'}(S) \geqslant -\sum_{\substack{s \in \mathcal{F} \\ v \in S}} P_{X,Y}(S) = \Delta_B P_{X,Y}(v)$ and

$\Delta_M P_{X,Y} = -(c-1)\Delta_B P_{X,Y}$.

### 3.4 PROOF OF THE SUFFICIENT CONDITION FOR THE BREAKER TO HAVE A WINNING STRATEGY

**Theorem 3.** *If $\mathcal{F}$ is a r-uniform hypergraph and $c \in \mathbb{N}$ satisfy $|\mathcal{F}| < c^{r-1}$, then Breaker has a winning strategy in the $G(\mathcal{F}, c)$.*

*Proof.* Let's define strategy for the Breaker: in each move $c - 1$ times pick such vertex $v$ whose choice reduces the potential of the position $P(X, Y)$ he most

$$y = \operatorname*{argmin}_{v \in \mathcal{V}(\mathcal{F}) \setminus (X \cup Y)} \Delta_B P_{X,Y}(v) = \left( = \operatorname*{argmax}_{v \in \mathcal{V}(\mathcal{F}) \setminus (X \cup Y)} |\Delta_b P_{X,y}(v)| \right)$$

, where $X$ and $Y$ — are the sets of points picked by Maker and Breaker respectively up to this point. We shall prove that this strategy guarantees winning for the Breaker.

To do this we shall prove, that in the end of the game the potential of the position is less than one. As mentioned in 3.1.2 it would imply that Breaker has won.

Let's split moves of players $\{x_1, y_1, \cdots y_{c-1}, x_2, y_c \ldots y_{2(c-1)} \ldots y_m\}$ into first move of Maker $\{x_1\}$, pairs of moves one by Breaker and one by Maker $\{y_{i(c-1)+1} \cdot y_{(i+1)(c-1)}, x_{i+1}\}$ and the last move of Breaker $\{y_i\}$ if it exists.

Let's see, how potential of the position changes as game progresses. Before players make any moves the potential is $P(\emptyset, \emptyset) = |\mathcal{F}| c^{-n} \leqslant c^{-2}$.

After first move $\{x_1\}$ the potential of each edge increases at most in $c$ times. Therefore after this move the potential of position $P(\{x_1\}, \emptyset) \leqslant cP(\emptyset, \emptyset) = c^{-1} < 1$.

Now, we prove that if Breaker follows the strategy, after one pair of moves $Y' = \{Y_1, Y_2 \ldots Y_{c-1}\}$ and $X' = \{x_{m+1}\}$ potential of the position does not increase. It would proof the theorem, since it shows that after all pairs of moves are played the potential of position is still less than 1. After last Breaker's move potential can't increase, therefore at the end of the game potential is less than 1. Thus Breaker wins if plays according to the strategy3.1.2.

Let's prove **?**. The change of the potential is

$$P\left(X \cup X', Y \cup Y'\right) - P\left(X, Y\right) = \tag{10}$$
$$= P\left(X \cup X', Y \cup Y'\right) - P\left(X, Y \cup Y'\right) + P\left(X, Y \cup Y'\right) - P\left(X, Y\right) = \tag{11}$$
$$= \Delta_M P_{X, Y \cup Y'}\left(x_{m+1}\right) + \underbrace{P\left(X, Y \cup Y'\right) - P\left(X, Y\right)}_{\text{term}} \leqslant \tag{12}$$

Since this term can be bounded
$$P\left(X, Y \cup \{Y_1, Y_2, \ldots Y_{c-1}\}\right) - P\left(X, Y\right) =$$

$$= \underbrace{\left(P\left(X, Y \cup \{Y_1, Y_2, \ldots Y_{c-1}\}\right) - P\left(X, Y \cup \{Y_1, Y_2, \ldots Y_{c-2}\}\right)\right)}_{=\Delta_B P_{X, Y \cup \{Y_1, Y_2, \ldots Y_{c-2}\}}(Y_{c-1})} +$$

$$+ \underbrace{\left(P\left(X, Y \cup \{Y_1, Y_2, \ldots Y_{c-2}\}\right) - P\left(X, Y \cup \{Y_1, Y_2, \ldots Y_{c-3}\}\right)\right)}_{=\Delta_B P_{X, Y \cup \{Y_1, Y_2, \ldots Y_{c-3}\}}(Y_{c-2})} +$$

$$+ \cdots +$$
$$+ \underbrace{\left(P\left(X, Y \cup \{Y_1\}\right) - P\left(X, Y\right)\right)}_{\Delta_B P_{X, Y \cup \emptyset}(Y_1)} =$$

$$= \Delta_B P_{X, Y \cup \{Y_1, Y_2, \ldots Y_{c-2}\}}\left(Y_{c-1}\right) + \Delta_B P_{X, Y \cup \{Y_1, Y_2, \ldots Y_{c-3}\}}\left(Y_{c-2}\right) + \cdots + \Delta_B P_{X, Y \cup \emptyset}\left(Y_1\right) \leqslant$$
[because $Y_i$ is selected to minimize this expression]
$$\leqslant \Delta_B P_{X, Y \cup \{Y_1, Y_2, \ldots Y_{c-2}\}}\left(Y_{c-1}\right) + \Delta_B P_{X, Y \cup \{Y_1, Y_2, \ldots Y_{c-3}\}}\left(Y_{c-1}\right) + \cdots + \Delta_B P_{X, Y \cup \emptyset}\left(Y_{c-1}\right) \leqslant$$
[because $Y \cup \{Y_1, Y_2, \ldots Y_i\} \subset Y \cup Y'$ and property 8]
$$\leqslant \Delta_B P_{X, Y \cup Y' \setminus \{Y_{c-1}\}}\left(Y_{c-1}\right) + \Delta_B P_{X, Y \cup Y' \setminus \{Y_{c-1}\}}\left(Y_{c-1}\right) + \cdots + \Delta_B P_{X, Y \cup Y' \setminus \{Y_{c-1}\}}\left(Y_{c-1}\right) =$$
$$= (c-1)\Delta_B P_{X, Y \cup Y'}\left(Y_{c-1}\right)$$

We can continue calculation 12 as

$$\leqslant \Delta_M P_{X, Y \cup Y'}\left(x_{m+1}\right) + (c-1)\Delta_B P_{X, Y \cup Y'(Y_{c-1})} \leqslant \text{[because of 9]} \leqslant$$
$$\leqslant \Delta_M P_{X, Y \cup Y' \setminus \{Y_{c-1}\}}\left(x_{m+1}\right) + (c-1)\Delta_B P_{X, Y \cup Y'(Y_{c-1})} = 0 \text{ [because of 7]}$$

$\square$

## 3.5 Sufficient condition for the Maker to have a winning strategy

**Theorem 4.** *If $\mathcal{F}$ is a $r$-uniform hypergraph and $c \in \mathbb{N}$ satisfy $\frac{|\mathcal{F}|}{|\mathcal{V}(\mathcal{F})|} > (c-1)^2 c^{r-3} d_2\left(\mathcal{F}\right)$ than the Maker has a winning strategy in $G\left(\mathcal{F}, c\right)$.*

*Proof.* Let's formulate the strategy for the Maker — to pick the vertex $x$ that increases the potential of position the most.
$$x = \operatorname*{argmax}_{x \in \mathcal{F} \setminus (X \cup Y)} \Delta_M P_{X, Y}\left(x\right)$$

In order to prove that this strategy guaranties win of the Maker, we shall prove, that it guarantees that in the end potential is positive [3.1.2]. In order to prove this let's split sequence of moves of players $\{x_1, y_1, \cdots y_c, x_2, y_{c+1} \ldots y_{2c}, \ldots y_m\}$ into pairs (one move of Maker and one move of Breaker) $x_i, \{y_{ic+1} \ldots y_{ic+c}\}$. The last pair may contain less than $c$ moves of breaker.

Let's see, how the potential of position changes as game progresses.

The potential of position in before the first move is $|\mathcal{F}| c^{-n} 3.1.1$

Let's prove that after one pair of moves the potential of the position decreases at most by $\left(1 - \frac{1}{c}\right)^2 d_2\left(\mathcal{F}\right)$, i.e. $P\left(X \cup \{x\}, Y \cup Y_{m+1}\right) - P\left(X, Y\right) \geqslant \left(1 - \frac{1}{c}\right)^2 d_2\left(\mathcal{F}\right)$.

This would prove the theorem since, after all $\frac{|\mathcal{F}|}{c}$ pairs of moves are played, the potential is at least

$$P\left(\emptyset,\emptyset\right)-\left(c-1\right)^{2}c^{-2}d_{2}\left(\mathcal{F}\right)\frac{\left|\mathcal{V}\left(\mathcal{F}\right)\right|}{c}=\left|\mathcal{F}\right|c^{-n}-\left(c-1\right)^{2}c^{-3}d_{2}\left(\mathcal{F}\right)\left|\mathcal{V}\left(\mathcal{F}\right)\right|$$

which is positive under the condition of the theorem. So, at the end of the game, potential is positive, and therefore Maker wins3.1.2.

Let's prove3.5. To do this we need to calculate the change of the potential after one pair of moves.

Informally if potential is $P\left(X,Y\right)$ on $m$-th move then after Maker's move the potential becomes $P\left(X\cup\{x\},Y\right)$ and after second player picks $c-1$ vertices $Y'=\{y_{1},y_{2},\ldots y_{c-1}\}$ the potential of the position becomes $P\left(X\cup\{x\},Y\cup\{y_{1}\}\right)$ then $P\left(X\cup\{x\},Y\cup\{y_{1},y_{2})\}$, ... then $P\left(X\cup\{x\},Y\cup\{y_{1},y_{2},\ldots y_{c-1}\}\right)$

Formally

$$P\left(X\cup\{x\},Y\cup Y_{m+1}\right)-P\left(X,Y\right)= \tag{13}$$
$$=\left[-P\left(X,Y\right)+P\left(X\cup\{x\},Y\right)\right]+ \tag{14}$$
$$+\left[-P\left(X\cup\{x\},Y\right)+P\left(X\cup\{x\},Y\cup\{y_{1}\}\right)\right]+ \tag{15}$$
$$+\left[-P\left(X,Y\cup\{y_{1}\}\right)+P\left(X\cup\{x\},Y\cup\{y_{1},y_{2}\}\right)\right]+ \tag{16}$$
$$+\cdots+ \tag{17}$$
$$+\left[-P\left(X,Y\cup\{y_{1},y_{2}\ldots y_{c-2}\}\right)+P\left(X\cup\{x\},Y\cup\{y_{1},y_{2}\ldots y_{c-1}\}\right)\right]= \tag{18}$$
$$=\Delta_{M}P_{X,Y}\left(x\right)+\Delta_{B}P_{X\cup\{x\},Y}\left(y_{1}\right)+\Delta_{B}P_{X\cup\{x\},Y\cup\{y_{1}\}}\left(y_{2}\right)+\Delta_{B}P_{X\cup\{x\},Y\cup\{y_{1},y_{2}\}}\left(y_{3}\right)+\ldots \tag{19}$$
$$\cdots+\Delta_{B}P_{X\cup\{x\},Y\cup\{y_{1},y_{2},\ldots y_{c-2}\}}\left(y_{c-1}\right)\geqslant \tag{20}$$
$$\geqslant\Delta_{M}P_{X,Y}\left(x\right)+\sum_{y\in Y'}\Delta_{B}P_{X\cup\{x\},Y}\left(y\right) \tag{21}$$

Since

$$\Delta_{B}P_{X\cup\{x\},Y}\left(y\right)=-\sum_{\substack{S\in\mathcal{F}\\y\in S}}P_{X\cup\{x\},Y}\left(S\right)=-\sum_{\substack{S\in\mathcal{F}\\y\in S\\x\notin S}}\underbrace{P_{X\cup\{x\},Y}\left(S\right)}_{\substack{=P_{X,Y}(S)\\\text{because }x\notin S}}-\sum_{\substack{S\in\mathcal{F}\\y\in S\\x\in S}}\underbrace{P_{X\cup\{x\},Y}\left(S\right)}_{\substack{=cP_{X,Y}(S)\\\text{because }x\in S}}= \tag{22}$$

$$=-\sum_{\substack{S\in\mathcal{F}\\y\in S\\x\notin S}}P_{X,Y}\left(S\right)-\sum_{\substack{S\in\mathcal{F}\\y\in S\\x\in S}}cP_{X,Y}\left(S\right)= \tag{23}$$

$$=\underbrace{-\sum_{\substack{S\in\mathcal{F}\\y\in S}}P_{X\cup\{x\},Y}\left(S\right)}_{=\Delta_{B}P_{X,Y}(y)}-\left(c-1\right)\sum_{\substack{S\in\mathcal{F}\\y\in S\\x\in S\text{ because }\overline{\{x,y\}}\subset S\setminus X}}\underbrace{P_{X,Y}\left(S\right)}_{\leq c^{-2}}\quad\geqslant \tag{24}$$

$$\geqslant\Delta_{B}P_{X,Y}\left(y\right)-\left(c-1\right)\left|\{S\in\mathcal{F}\mid\{x,y\}\subset S\}\right|c^{-2}\geq \tag{25}$$

$$\geqslant\Delta_{B}P_{X,Y}\left(y\right)-\left(c-1\right)c^{-2}d_{2}\left(\mathcal{F}\right) \tag{26}$$

The calculation21 can be continued as

$$P\left(X \cup \{x\}, Y \cup Y_{m+1}\right) - P\left(X, Y\right) = \tag{27}$$

$$\geqslant \Delta_M P_{X,Y}\left(x\right) + \sum_{y \in Y'} \Delta_B P_{X \cup \{x\}, Y}\left(y\right) \geq \tag{28}$$

$$\geqslant \Delta_M P_{X,Y}\left(x\right) + \sum_{y \in Y'} \Delta_B P_{X,Y}\left(y\right) - \sum_{y \in Y'} \left(c-1\right) c^{-2} d_2\left(\mathcal{F}\right) \geqslant \tag{29}$$

$$\geqslant \underbrace{\Delta_M P_{X,Y}\left(x\right)}_{\substack{= \max\limits_{x \in \mathcal{F} \backslash (X \cup Y)} \Delta_M P_{X,Y} \\ \text{in Makers strategy} \\ \text{move } x \text{ is chosen} \\ \text{to maximize this}}} - \frac{1}{c-1} \sum_{y \in Y'} \Delta_M P_{X,Y}\left(y\right) - \sum_{y \in Y'} \left(c-1\right) c^{-2} d_2\left(\mathcal{F}\right) \geq \tag{30}$$

$$\geqslant \underbrace{\max_{x \in \mathcal{F} \backslash (X \cup Y)} \Delta_M P_{X,Y} - \frac{1}{|Y'|} \sum_{y \in Y'} \Delta_M P_{X,Y}\left(y\right)}_{\geqslant 0} - \left(c-1\right) \left(c-1\right) c^{-2} d_2\left(\mathcal{F}\right) \geqslant \tag{31}$$

$$\geqslant \left(c-1\right)^2 c^{-2} d_2\left(\mathcal{F}\right) \tag{32}$$

Thus, if Maker plays according to the strategy (of picking the move that has maximize the $\Delta_M P_{X,Y}$), then after one pair of moves the potential can decrease only by $\left(c-1\right)^2 c^{-2} d_2\left(\mathcal{F}\right)$.

$\square$

## 4  GAMES ASSOCIATED WITH CLASSICAL PROBLEMS AND THEIR VARIATIONS

Firstly, we'll apply those theorems to study game forms of classic Van der Waerden problems and their natural variations

### 4.1  VANDER VAN DER GAME WITH $c$ COLORS

Van der Wander Game [Beck (1981)] with $k$ colors is defined as Maker-Breaker game corresponding to the hypergraph, with set of vertexes $S_N = \{1, 2, \ldots, N\}$ and set of edges $\mathcal{F}$ consisting of all arithmetic progressions of length $d$ ($d$ is a parameter of the game).

#### 4.1.1  CONDITION FOR THE MAKER TO HAVE A WINNING STRATEGY

**Theorem 5.** *If $N > 2\left(n-1\right)^2 n \left(c-1\right)^2 c^{n-3}$ then Maker has a winning strategy.*

*Proof.* Let's check that sufficient conditions for a Maker to have a winning strategy are met.

1. By definition each edge of $\mathcal{F}$ contains exactly $n$ elements, so $\mathcal{F}$ is $n$-uniform

2. $|\mathcal{V}\left(\mathcal{F}\right)| = |\{1, 2 \ldots N\}| = N$

3. For every number $b < \frac{N}{2}$ for every step $s < \frac{N}{2(n-1)}$ the arithmetic progression $b, b + s, b + 2s, \ldots b + (n-1) s$ is contained in $\mathcal{V}\left(\mathcal{F}\right)$ (and two different progressions cannot have both same beginnings and steps). Because of that the number of edges is at least $\frac{N}{2} \frac{N}{2(n-1)} = \frac{N^2}{4(n-1)}$

4. By two numbers $a$ and $b$ and their number in arithmetic progression the can be unambiguously determined. Because of that for given two numbers $a$ and $b$ the number of arithmetic progressions containing both of them is at most the number of pairs of indexes, which is $\binom{n}{2} = \frac{n(n-1)}{2}$

Therefore $\frac{|F|}{|\mathcal{V}(\mathcal{F})|} \geqslant \frac{N}{4(n-1)} \geqslant \left(c-1\right)^2 c^{n-3} \frac{n(n-1)}{2} = \left(c-1\right)^2 c^{n-3} d_2\left(\mathcal{F}\right)$, so the condition for Maker to have a winning strategy is met.

$\square$

### 4.1.2 CONDITION FOR THE BREAKER TO HAVE A WINNING STRATEGY

**Theorem 6.** *If $N < \sqrt{(n-1)\, c^{n-1}}$ then Breaker has a winning strategy.*

*Proof.* Let's check that sufficient conditions for a Breaker to have a winning strategy are met.

1. $\mathcal{F}$ is $n$-uniform.

2. Since for every first element of progression ($N$ options to choose) and for each step (at most $\frac{N}{n-1}$ options to choose) there can be only one progression with this start and step (and all progressions are counted that way) $|F| \leq \frac{N^2}{n-1}$.

So, $|F| = \frac{N^2}{n-1} < \frac{(n-1)c^{n-1}}{n-1} = c^{n-1}$ $\qquad\qquad\square$

Therefore we can establish bounds on minimal $N(n, c)$ which is enough for the Maker to have a winning strategy. $N(n, c)$ between $\sqrt{(n-1)\, c^{n-1}}$ and $2n\,(n-1)\,(c-1)^2\, c^{n-3}$.

## 4.2 $d$-DIMENSIONAL VAN DER WAERDEN GAME

Here the game is defined by numbers $N$, $d$, amount of colors $c$ and figure $A \subset \mathbb{R}^d$. The set of vertices is $N$-dimensional cube $\{1, 2, \dots N\}^d$ and set of vertexes is $v + \lambda A$, where $v \in \mathbb{R}^d$ and $\lambda \in \mathbb{N}$. Van der Waerden Game is a specific case where $d = 1$ and $A = \{1, 2 \dots n\}$.

Despite more complex structure lower and upper bound can be obtained using similiar methods.

### 4.2.1 CONDITION FOR THE MAKER TO HAVE A WINNING STRATEGY

**Theorem 7.** *If $N > n\,(n-1)\, 2^d diam\,(A)\,(c-1)^2\, c^{n-3}$, where $diam\,(A)$ is minimal size of cube that contains A, maker has a winning strategy.*

*Proof.* Let's check characteristics of the hypergraph $\mathcal{F}$

1. $\mathcal{F}$ is $n$-uniform.

2. $|\mathcal{V}(\mathcal{F})| = \left|\{1, 2, \dots N\}^d\right| = N^d$

3. For every $v \in \left|\{1, 2, \dots \frac{N}{2}\}\right|$ and every $\alpha \in \{1, 2, \dots \frac{N}{2 diam(A)}\}$ gives a unique figure $v + \alpha A$ that does not exceed the boundaries of $\{1, 2, \dots N\}^d$. So, the amount of figures is at least $\left(\frac{N}{2}\right)^d \frac{N}{2 diam(A)} = \frac{N^{d+1}}{2^{d+1} diam(A)}$.

4. Let's calculate $d_2(\mathcal{F})$ — how many edges $v + \alpha A$ can contain two points $P_1$ and $P_2$. If we know, what points $A_i$ and $A_j$ of $A$ are mapped to the $P_1$ and $P_2$ respectively, than there is at most one solution of $(v + \alpha A_i = P_1) \wedge (v + \alpha A_j = P_2)$, since $\alpha\,(A_i - A_j) = P_1 - P_2$ have at most one solution for $\alpha$ and then $v = P_1 - \alpha A_i$ have only one solution for $v$. Therefore, amount of edges $v + \alpha A$, such that $\{P_1, P_2\} \subset v + \alpha A$ is at most amount of pairs $A_i, A_j$ i.e. $\binom{2}{n} = \frac{n(n-1)}{2}$.

So $\frac{|F|}{|\mathcal{V}(\mathcal{F})|} \geqslant \frac{N^{d+1}}{2^{d+1} diam(A)} \frac{1}{N^d} = \frac{N}{2^{d+1} diam(A)} > (c-1)^2\, c^{n-3} \binom{2}{n}$ under the condition of the theorem. So the condition for the Maker to have a winning strategy is met. $\qquad\square$

### 4.2.2 CONDITION FOR THE BREAKER TO HAVE A WINNING STRATEGY

**Theorem 8.** *If $N < \sqrt[d+1]{diam\,(A)\, c^{n-1}}$, than Breaker has a winning strategy.*

*Proof.* Let's check characteristics of the hypergraph $\mathcal{F}$

- $\mathcal{F}$ is $n$-uniform.

- $|F| \leq N^{d+1}$, since there is $N^d$ options for $v$ and $\frac{N}{\text{diam}(A)}$ for $\alpha$

So, $|F| = \frac{1}{\text{diam}(A)} N^{d+1} < c^{n-1}$ under conditions of the theorem. $\qquad \square$

## 4.3 HALES AND JEWETT GAME

Hales and Jewett game is similar to the $n$-dimensional Van der Waerden Game, but instead of increasing length of $d$-dimensional cube, we increase dimensionality of $d$ of $\{1, 2 \ldots n\}^d$. Maker's goal is to occupy non-constant arithmetic progression containing $n$ points. Here arithmetic progression is the set of type $\{a + ib \mid 1 \leq i \leq n\}$ for some point $a$ and non-zero vector $b$.

**Theorem 9.** $|\mathcal{V}(\mathcal{F})| = \frac{1}{2}\left((n+2)^d - n^d\right)$

*Proof.* For every arithmetic progression can be defined using start point $a$ and step vector $b$ as $\{a + ib \mid 1 \leq i \leq n\}$, where $a$ is start point and nonzero $b$ is the step vector. For every line, there is exactly two pairs of $(a, b)$ that generate the line. (one defines line "from start to finish" and one "from finish to start"). So, in order to calculate amount of combinatorial lines, we can multiply $\frac{1}{2}$ and the amount of pairs of pairs $(a, b), b \neq 0$ for witch no points of type $a + ib, 1 \leq i \leq n$ does go over bounderies of the cube.

To calculate the amount of those pairs, let's look at coordinates of $a$ and $b$ to an $k$-th axis. It is progression arithmetic $a_k + ib_k$, which fits in $\{1, 2, \ldots, N\}$. There is exactly $n + 2$ such arithmetic progressions of length $n$: $n$ constant ones and $2$ containing all elements. Because we can select coordinates in each of $d$ axis independently, there are in total $(n+2)^d$ pairs of $(a, b)$ which generate arithmetic progression of length $n$ which is contained inside the cube $\{1, 2, \ldots, n\}^d$. From those ones there is $n^d$ ones with vector $b = 0$. Thus only $(n+2)^d - (n)^d$ pairs generate non-constant arithmetic progression that fits in $(1, 2, \ldots, n)^d$. Therefore the amount of sets of arithmetic progressions is $|\mathcal{V}(\mathcal{F})| = \frac{1}{2}\left((n+2)^d - n^d\right)$ $\qquad \square$

### 4.3.1 CONDITION FOR THE MAKER TO HAVE A WINNING STRATEGY

**Theorem 10.** *If* $d > \log 2 (c-1)^2 c^{n-3} + 1 - \log 1 + \frac{2}{n}$, *Maker has a winning strategy.*

*Proof.* Let's check characteristics of the hypergraph $\mathcal{F}$

- By definition of $\mathcal{F}$, every edge contains $n$ vertices, so $\mathcal{F}$ is $n$-uniform

- $|\mathcal{V}(\mathcal{F})| = \left|\{1, 2, \ldots n\}^d\right| = n^d$

- $|\mathcal{F}| = \frac{1}{2}\left((n+2)^d - n^d\right)$

- Because standard embedding of the cube in $\mathbb{R}^d$ is injection that transforms combinatorial lines into $n$ points that lie on a geometrical line in $\mathbb{R}^d$ and two points in $R^d$ uniquely define a geometrical line, two points uniquely define combinatorial line. So $d_2(F) = 1$.

$\qquad \square$

So $\frac{|\mathcal{F}|}{|F|} = \frac{1}{2}\left(\left(1 + \frac{2}{n}\right)^d - 1\right) > (c-1)^2 c^{n-3}$ under the condition of theorem.

### 4.3.2 Condition for the Breaker to have a winning strategy

**Theorem 11.** *If $d < (n-1)\log c - \log n + 2$, then Breaker has a winning strategy.*

*Proof.* Let's check characteristics of the hypergraph $\mathcal{F}$

- $\mathcal{F}$ is $n$-uniform.
- $|\mathcal{F}| = \frac{1}{2}\left((n+2)^d - n^d\right) \leq (n+2)^d$.

Therefore $|\mathcal{F}| < (n+2)^d < c^{n-1}$ $\qquad\qquad\qquad\qquad\qquad\square$

## 5 Another family of Ramsey type problems: games on $\mathbb{R}^d$

Another interesting family of problems in Ramsey theory studies colorings of $\mathbb{R}^d$:

**Statement 1.** *The $\mathbb{R}^d$ is colored into $c$ colors and there is finite figure $F$. Is it guaranteed that exists monocromatic (all vertices have the same color) $F'$ that can be obtained as $F' = v + g_\alpha(F)$), where $v+$ is translation to the vector $v$ and $g_\alpha$, $\alpha \in \mathbb{R}^m$ is some transformation.*

This family includes several popular problems

- "Is it true, that for every coloring of $\mathbb{R}^d$ exist monochromatic geometrical copy of $A$?". Here $g_\alpha(x) = \alpha x$, where $\alpha \in O_d$. In this case simple example shows that Ramsey statement is false.
- "Is it true, that for every coloring of $\mathbb{R}^d$ exist monochromatic "hyperbolic" copy version of $A$. Here by "hyperbolic copy" we mean $g_\alpha A$, where $\alpha \in \left\{ \{\alpha_1, \alpha_2, \ldots \alpha_d\} \in \mathbb{R}^d \mid \prod_{i=1}^{n} \alpha_i = 1 \right\}$ and $g_\alpha(\{x_0, x_1, \ldots x_d\}) = (\alpha_0 x_0, \alpha_1 x_1, \ldots, \alpha_d, x_d)$. This problem in general case is still open, and best known result is that Ramsey statement of this is true if $A$ is a simplex Sharich et al. (2021).
- "Is it true that for every coloring of $\mathbb{R}^d$ exists monochromatic figure homothetical to $A$". Here $\alpha \in \mathbb{R}$, and $g_\alpha(x) = \alpha x$. This is continuous variation of the Van den Waerden theorem.

### 5.1 Game associated with those problems

Let $A = \{a_0, a_1, \ldots, a_n\} \subset U$ — where $U$ is an open setset of $\mathbb{R}^d$, $n \geqslant 2$ and $F_\alpha$, $\alpha \in M$ - is family of functions $\mathbb{R}^d \to \mathbb{R}^d$. As usual we define a Maker-Breaker game for $c$ colors $G(A, U, F, c, M)$, where Maker and Breaker alternately pick points on $U$ and Maker wins if he occupies a figure of type $F_\alpha(A)$ for some $\alpha \in M$.

**Theorem 12.** *If $\{F_\alpha, \alpha \in M\}$ is a family of functions that contains a subfamily $\{F_\alpha, \alpha \in M' \subset M\}$ such that*

- *$F$ is continuous by both $\alpha$ and $v$ and exists $\alpha_0$, such that $F_{\alpha_0} = Id$.*

- *For every $\alpha_1, \alpha_2 \in M'$ holds $F_{\alpha_1}(A) \neq F_{\alpha_2}(A)$.*

- *Exists $n \in \mathbb{N}$ such that for every $u, u' \in U$ and $a, a' \in A$, $\{\{v, \alpha\} \mid (v + F_\alpha(a) = u) \wedge (v + F_\alpha(a') = u')\}$ consist of less than $n$ elements.*

- *Set of $\alpha$ for which $F_\alpha(A) \subset \mathbb{Q}(A)$ is dense in $M'$.*

*Then Maker has a winning strategy.*

Note, that these conditions does not have any restrictions on $U$, size of $A$ or amount of colors $c$ and includes game forms of all of cases from the above.

*Proof.* Consider $B_\epsilon$ is a ball of radius $\epsilon$, $C_\epsilon = A + B_\epsilon$ — set of balls around each point of $A$ and $\mathbb{Q}_n = \{\frac{p}{q} \mid p \in \mathbb{Z}, q \in \{1, 2, \ldots n\}\}$ and $\mathbb{Q}_{n,A} = \{q_1 a_1 + \ldots q_n a_n \mid a_i \in A, q_i \in \mathbb{Q}_n\}$ and $\mathbb{Q}_{n,\epsilon} = \mathbb{Q}_n \cap C_\epsilon$. We shall prove, that if we restrict set of vertices to the $\mathbb{Q}_{n,\epsilon}$ some $n$ and $\epsilon$, then condition for a Maker to have a winning strategy is met. Therefore he has a strategy to always win using subset of all set of vertices.

Let's restrict all functions domains to $U$. Since $F_\alpha$ is continious on $\alpha$, there exists neighbourhood $O_\epsilon$ of $\alpha_0$, such that $\left| F_\alpha - \underbrace{F_{\alpha_0}}_{Id} \right| < \epsilon$. Because the set of $\alpha$ for which $F_\alpha(A) \in \mathbb{Q}(A)$ is dense in $M'$ there are infinitely many $\alpha$ in $O_\epsilon$, such that $F_\alpha(A) \subset \mathbb{Q}(A)$ in addition to $|F_\alpha - Id| < \epsilon$, which implies $F_\alpha(A) \subset C_\epsilon \cap \mathbb{Q}_n$. Let's denote the set of those $\alpha$-s as $O'_\epsilon$.

Since $\lim_{i \to \infty} \mathbb{Q}_i = \mathbb{Q}$ we know that $\lim_{n \to \infty} \mathbb{Q}_{n,\epsilon} = \mathbb{Q}(A) \cap C_\epsilon$. Therefore amount of $\alpha$ such that $F_\alpha(A) \in Q_{n,\epsilon}$ approaches to infinity as $n$ grows. Let's denote those alphas as $O'_{n,\epsilon}$, $\lim_{n \to \infty} |O'_{n,\epsilon}| = |O'_\epsilon| = \infty$

Finally, let's check the conditions for Maker to have a winning strategy. Here $\mathcal{F} = \{v + F_\alpha A \mid \alpha \in O'_{n,\epsilon/2}, v \in \mathbb{Q}_{n,\epsilon/2}\}$ - hypergraph of sets for Maker to win. Since both $f(x) = v + x$ and $F_\alpha$ under the conditions of the theorem move point to the distance at most $\epsilon/2$, we can say $\mathcal{V}(\mathcal{F}) = \mathbb{Q}_{n,\epsilon}$.

- For $\epsilon$ small enough, $|F_\alpha - Id| < \epsilon$ implies that $F_\alpha A$ contains $|A|$ elements. Because $v + \cdot$ is bijection every element of $\mathcal{F}$ contains $|A|$ elements. So $\mathcal{F}$ is $|A|$-uniform.

- $|\mathcal{V}(\mathcal{F})| = |\mathbb{Q}_{n,\epsilon}| = 2^d |\mathbb{Q}_{n,\epsilon/2}| + o(|\mathbb{Q}_{n,\epsilon/2}|)$. The last is property of balls in $\mathbb{Q}^n$.

- From definition of $\mathcal{F}$ we can conclude $|F| = |O'_{n,\epsilon/2}| \cdot |\mathbb{Q}_{n,\epsilon/2}|$.

- $d_2(\mathcal{F}) < D$ is reformulation of "There exists $n \in \mathbb{N}$ that for every $u, u' \in U$ and $a, a' \in A$, $\{\{v, \alpha\} \mid (v + F_\alpha(a) = u) \wedge (v + F_\alpha(a') = u')\}$ consist of less than $n$ elements." from conditions of theorem.

Therefore $\frac{|F|}{|\mathcal{V}(\mathcal{F})|} = \frac{|O'_{n,\epsilon/2}| \cdot |\mathbb{Q}_{n,\epsilon/2}|}{2^d |\mathbb{Q}_{n,\epsilon/2}| + o(|\mathbb{Q}_{n,\epsilon/2}|)} = \frac{1}{2^d} |O'_{n,\epsilon/2}| + o(1) \to \infty$ if $n \to \infty$. Hence exists $n$ such that $\frac{|F|}{|\mathcal{V}(\mathcal{F})|} > (c-1)^2 c^{n-3} d_2(F)$. $\square$

Let's show that this theorem can be applied game form of each from this three problems.

## 5.2 Game form of "Existing of hyperbolical copy of $A$" for $d \geqslant 2$

Note: we prove for the case, where every $a_i - a_j$ for $i \neq j$ is not parallel to any axis. In future we want to remove this condition from the theorem.

Here figure $A$ is fixed, $M = \{\{\alpha_1, \ldots \alpha_n\} \mid \prod \alpha_i = 1\} = H^n$ - hyperboloid and transformation is $g_\alpha(\{x_0, x_1, \ldots x_d\}) = (\alpha_0 x_0, \alpha_1 x_1, \ldots, \alpha_d, x_d)$.

Let's define $M' = M$. Let's check that $M'$ satisfies requirements of the theorem above.

- $F$ is continuous on $\alpha$ and $u$.
- Let's prove that the set $\{\{v, \alpha\} \mid (v + F_\alpha(a) = u) \wedge (v + F_\alpha(a') = u')\} = \{\{v, \alpha\} \mid (v + \alpha \circledast a = u) \wedge (v + \alpha \circledast a' = u')\} = \{\{v, \alpha\} \mid (\alpha \circledast (a - a') = u - u') \wedge (v = u - \alpha \circledast a)\}$ which can contain contain at most 1 element.
- Since for every $\alpha \in \mathbb{Q}^d$ $F_\alpha(A) = \{\alpha a \mid a \in A\} \subset \mathbb{Q}(A)$ and $\mathbb{Q}^d$ is dense everywhere in $M\mathbb{R}^d$ if $d \geqslant 2$.

So, the condition are met. Therefore Maker has a winning strategy.

## 5.3 Continious Van der Waerden game

Here $M = \mathbb{R}$, $F_\alpha(x) = \alpha x$. Let's take $M' = M$, then

- $F$ is continious by $\alpha$ and $x$
- For every $u, u' \in U$ and $a, a' \in A$, the set $\{\{v, \alpha\} \mid (v + F_\alpha(a) = u) \wedge (v + F_\alpha(a') = u')\} = \{\{v, \alpha\} \mid (v + \alpha a = u) \wedge (v + \alpha a' = u')\} = \{\{v, \alpha\} \mid (\alpha(a - a') = u - u') \wedge (v = u - \alpha a)\}$ which can contain at most one element.
- For every $\alpha \in \mathbb{Q} \subset \mathbb{R} = M$ $F_\alpha(A) = \alpha A \subset \mathbb{Q}(A)$

So all condition for theorem above are met, therefore Maker can force a win.

## 5.4 Game form of "Existing of geometrical copy of $A$" for $d > 2$

Here $M = SO_n$. Let's select $M'_\alpha$, $\alpha \in S^1$ — family of all rotations $\rho_\alpha$ around a specific plane. Let's extend $A$ by adding $A^\perp = \rho_{\pi/2}(A)$. Let's check that all conditions for Maker to have a winning strategy are met.

- $F$ is continious by $\alpha$ and $x$.
- Let's prove that the set $\{\{v, \alpha\} \mid (v + F_\alpha(a) = u) \wedge (v + F_\alpha(a') = u')\}$ can contain at most one element. Indeed if $v + \rho_\alpha(a) = u$ and $v + \rho_\alpha(a') = u'$, then $\rho_\alpha(a - a') = u - u'$, but that means that $\alpha$ must be angle between $a - a'$ and $u - u'$, so there is only one option for $\alpha$, and since $v = u - \rho_\alpha v$ there is at most one option for $v$.
- It is known property of $S^1$ that $S^1 \cap \mathbb{Q}^2$ is dense in $S^1$. Every of those $\alpha$ produces transform that can be written as $\rho_\alpha(a) = q_1 a + q_2 a^\perp$, where $q_1$ and $q_2$ are rationals, so $\rho_\alpha(A \cup A^\perp) \subset \mathbb{Q}(A \cup A^\perp)$

## 6 Conclusion

In this paper we studied one approach to tackle Ramsey type problems by studying Maker-Breaker games associated with them. We found that greedy strategies work surprisingly well in this case and on some conditions are the winning ones and they solve games in $\mathbb{R}^n$ almost entirely. In other they can provide good estimates of who is winning or be a starting point to construct more complex or game-specific strategy.

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
