# OpenReview forum: "Game forms of Ramsey type problems."
_mathai.club/MathAI/2025/Conference — MathAI 2025 Oral_

### Official Review · Reviewer_F3wX · 2025-02-25
**Good article with good math but needs some improvements.**

**Rating:** 6
**Confidence:** 4

**Review:**

Strengths:
The article is written clearly and understandably. With good mathematics and correct formulas.
Solving problems using game theory is a very frequently used section of mathematics. Such approaches are always interesting.

Weaknesses:

1. There are some corrections both in English and in mathematical formulas (lines 152, 186).
2. It is necessary to more clearly reformulate the problem being solved (lines 21 and 31) and provide references to the literature
3. Take the formulations of lemmas and their proofs out of the proof of theorems.

Recommendations (mandatory):
Make a section on current research on the application of game theory in artificial intelligence. Also look at how LLM solves problems of this kind, as well as other practical implementations. Also make references to literature

---

### Official Review · Reviewer_RqxM · 2025-02-25
**The article is good, but does not fit the theme of the conference**

**Rating:** 5
**Confidence:** 3

**Review:**

This paper is all about research in the field of Ramsey theory, which is a section of combinatorics and studies the conditions under which a certain order necessarily arises in arbitrarily generated mathematical structures.One of the approaches to solving such problems is to introduce games related to these problems and then analyse them.This paper considers one of the methods for solving Ramsey problems based on the Maker-Braker game, and it presents several theorems demonstrating the existence of sufficient conditions for the existence of winning strategies.

The paper is well structured, written in clear mathematical language. However, the paper does not indicate the actuality of this research at all. In particular, the list of literature mainly consists of publications published in the last century, which also raises questions about the actuality of this research.  We would also like to note that the paper does not indicate the applicability of this research to artificial intelligence tasks.

So, it's a presentation of mathematical research in combinatorics and game theory, which is great, but I'm not sure it's the best fit for conference ‘Mathematics of Artificial Intelligence’.

---

### Official Review · Reviewer_6gr3 · 2025-02-27
**Hard to justify the relevance to the conference topic**

**Rating:** 6
**Confidence:** 4

**Review:**

The mathematical content is interesting. However, it is hard to justify the relevance to the main topic of the conference (AI).

---

### Decision · Program_Chairs · 2025-03-08

**Decision:**

Accept (Oral)

**Comment:**

Your article has been accepted and you can make a presentation on the article. All articles will be sorted by rating and within the available conference places one author from each article will be invited. If there are not enough places, then you will either have the opportunity to present remotely or come at your own expense!